# Quantitative earthquake-like statistical properties of the flow of soft materials below yield stress

P.K. Bera [1], S. Majumdar[2], G. Ouillon[3], D. Sornette[4,5] & A.K. Sood [1]*

The flow behavior of soft materials below the yield stress can be rich and is not fully understood. Here, we report shear-stress-induced reorganization of three-dimensional solid-like soft materials formed by closely packed nematic domains of surfactant micelles and a repulsive Wigner glass formed by anisotropic clay nano-discs having ionic interactions. The creep response of both the systems below the yield stress results in angular velocity fluctuations of the shearing plate showing large temporal burst-like events that resemble seismic foreshocks-aftershocks data measuring the ground motion during earthquake avalanches. We find that the statistical properties of the quake events inside such a burst map on to the scaling relations for magnitude and frequency distribution of earthquakes, given by Gutenberg-Richter and Omori laws, and follow a power-law distribution of the inter-occurrence waiting time. In situ polarized optical microscopy reveals that during these events the system self-organizes to a much stronger solid-like state.

[1] Department of Physics, Indian Institute of Science, Bangalore, Karnataka 560012, India. [2] Raman Research Institute, Bangalore, Karnataka 560080, India. [3] Lithophyse, 4 rue de l'Ancien Sénat, 06300 Nice, France. [4] D-MTEC, and Department Physics and Department of Earth Sciences, ETH Zürich, Scheuzerstrasse 7, CH-8092 Zürich, Switzerland. [5] Institute of Risk Analysis Prediction and Management, Academy for Advanced Interdisciplinary Studies, Southern University of Science and Technology, Shenzhen 518055, China. *email: asood@iisc.ac.in

Earthquakes, complex reorganization of earth crust caused mainly by the sudden release in energy from the sliding of geological faults, are probably the most severe natural phenomena to adversely affect human lives. To cut through the complexity of such a large-scale reorganization phenomenon, there have been significant recent efforts in mimicking earthquakes in controlled laboratory experiments by studying the deformation and failure in various solid materials under external loads. These include fracture of rock samples under compression[1–6], fracture of artificial rock of sintered polystyrene beads[7], compression of mesoporous silica ceramics[8], avalanches in wood compression[9] and stick–slip instabilities under shear in a two-dimensional assembly of polymer disks[10]. A recent study[11] demonstrates that charcoal samples damped with ethanol show avalanche events similar to earthquakes due to the internal stresses generated from ethanol evaporation. In all experiments on three-dimensional systems, the readout of the events is acoustic emission in the form of crackling noise caused by predominant irreversible deformations of the optically opaque solid samples with typical shear moduli ∼MPa or higher[12,13]. This raises the question of upscaling, i.e., how laws at the laboratory scale translate at the geological scale of a tens to hundreds of kilometers. Moreover, all these experiments are performed with a very high sampling rate (∼100 KHz and above), whereas seismic vibrations of engineering significance occur at frequencies from <0.2 Hz to 20 Hz[14]. Furthermore, the Poisson ratio for the rock samples, porous materials, and soft wood, being significantly less than 0.5[15,16], there is an overall volume contraction of the materials, which is counterfactual to earthquake effects on the Earth crust: most earthquakes are indeed double couples and thus conserve volume. Recent numerical simulation study of creep response of a yield-stress system of bubbles[17] indicates the possibility of observing earthquake-like statistics in soft matter systems without the need of ad hoc friction. There are a few reports of Gutenberg–Richter-like scaling laws—configurational free entropy changes in colloidal glass due to reorganizational events[18]; probability distribution of the force drop in stick–slip motion between two polymer plates[19] and stress fluctuation in polymer network[20]. Despite these experimental and simulation studies, to our knowledge, there is no experimental study on soft, continuous, and disordered materials with solid-like yield stress of few Pa, that reports simultaneously all the three well-known scaling relations and statistics (discussed below) similar to earthquake avalanches or controlled laboratory experiments mentioned above. Such study is important, since their mechanical properties at the laboratory scale may be more adapted as analogs of the geological scales. Moreover, the mesoscopic domain structures in this broad class of materials can be easily probed, and can be reversibly controlled by an applied shear stress or an external field[21]. In general, similar tunability of domain structures is not accessible for conventional solids. In very few cases, the stress-induced evolution of domain structures have been observed in solids[13], however, how the structural evolution change the overall shear modulus of the system is not clear. The major challenge to study reorganization events in soft yield-stress materials originates from the fact that they are much softer (shear moduli ∼tens of Pa, Supplementary Fig. 1) compared with those used in experiments mentioned earlier. Materials with such low values of shear moduli do not produce audible crackling noise under stress and, therefore, the statistical properties of reorganization events cannot be read out using acoustic emissions.

Here, using shear rheology along with polarized optical microscopy, we study the statistical properties of stress-induced reorganization in soft yield-stress materials. Accurate determination of the value of yield stress in soft and disordered materials is very challenging[22–24]. Recent experiments on a wide variety of soft materials[25] confirm the existence of a solid-like state below the yield stress consistent with the Herschel–Bulkley (HB) model. In our study, the estimated values (obtained by fitting the HB model to the experimental data) of the yield stress are of the order of a few tens of Pa (Fig. 1a). We apply perturbations in the form of shear stress and, since the system is incompressible, there is no volume strain in the sample. We directly measure and analyze the mechanical response of the sample by measuring the angular displacement ($\phi$) of the shearing plate with a time resolution of 0.04 s. For constant stress well below the yield stress, the angular velocity ($\dot{\phi}$) of the top plate (proportional to the shear rate) shows rapid temporal fluctuations around zero[26,27]. For the first time we

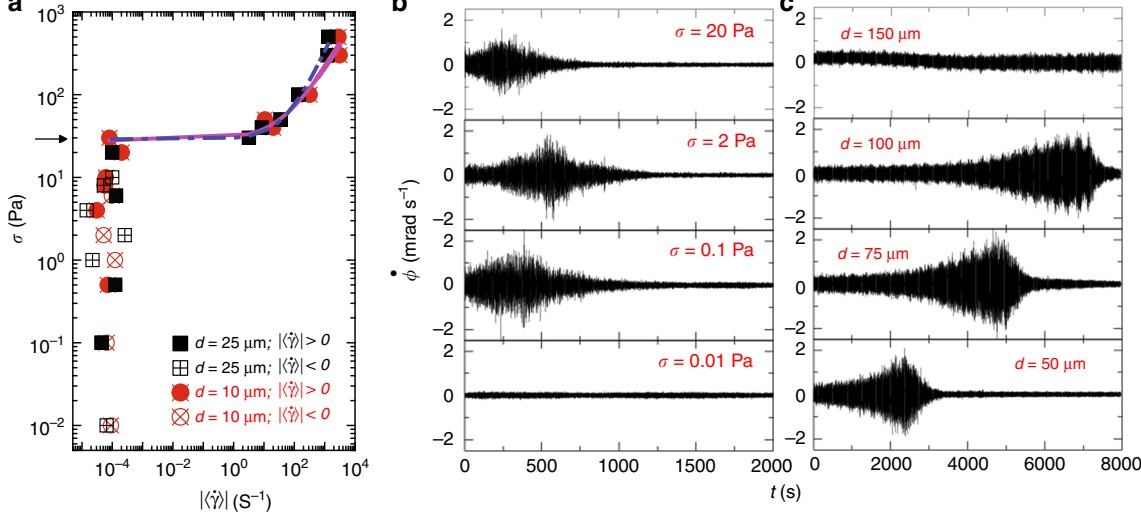

**Fig. 1 Steady-state flow curves and creep behavior of CTAT nematic. a** Applied shear stress ($\sigma$) versus absolute value of steady-state shear rate ($|\langle\dot{\gamma}\rangle|$, calculated considering data from 2000 s to 8000 s for each stress value) for 39 wt% CTAT + water sample at 30 °C (nematic phase of cylindrical micelles) is plotted for two different sample thicknesses ($d$). Filled symbols indicate positive and empty symbols indicate negative values of $\langle\dot{\gamma}\rangle$. Solid curve and dash-dot curve are Herschel–Bulkley fits with yield stress $\sigma_y = 28$ Pa, 29 Pa (indicated by the arrow) and power $n = 0.62, 0.81$ for $d = 10$ μm, 25 μm, respectively. **b** Angular velocity of the top plate ($\dot{\phi}$) as a function of time ($t$) is plotted for four different $\sigma$ with same sample thickness $d = 10$ μm (see Supplementary Fig. 3 for the expanded version of the $\sigma = 2$ Pa data), and **c** shows plots for four different $d$ with same applied stress $\sigma = 2$ Pa.

find that, for small sample thicknesses (the gap $d$ between shearing plates of a few tens of microns), the angular velocity fluctuations show burst-like events (Fig. 1b, c) persisting over about thousands of seconds (comparable with seismic foreshock to aftershock timescale), arising from slow building up and decay of angular velocity amplitudes. Such burst-like events closely resemble seismograph data during earthquakes resulting from failure and reorganization events of the earth crust over large length scales. In our system, these events indicate stress-induced reorganizations in the sample below yielding. Remarkably, the statistics of angular velocity fluctuations during these reorganization events follow well-known scaling relations given by Gutenberg–Richter law[28], the direct and inverse Omori laws[29] and show power-law distributions of inter-occurrence waiting time[30], qualitatively similar to earthquake avalanches.

In this study, we describe the creep behavior of two well-characterized, soft yield-stress materials: a dense nematic phase of rod-like surfactant micelles[31] with micron size randomly oriented nematic domains which are easily visible using polarized optical microscopy (POM), and a repulsive Wigner glass formed by a colloidal suspension of nanometer-sized laponite platelets[32]. The glassy state of laponite suspensions is formed due to long-range Coulomb repulsion between the charged platelets. Due to small domain structure, laponite system cannot be probed using POM. We discuss about the first system in detail and also touch upon the second one at the end.

## Results

**Burst phenomena in creep flow of CTAT nematic**. Figure 1a shows the steady-state flow curves (shear stress vs shear rate) of the nematic phase formed by 39 wt% of surfactant Cetyl-trimethylammonium Tosylate (CTAT) in water (see the Methods section). Below the yield point, the steady-state shear rate $\dot{\gamma} = \frac{R}{d} \times \dot{\phi}$ ($R$ is the radius of the top plate) shows positive and negative values with a very small average magnitude $|\langle\dot{\gamma}\rangle| \sim 10^{-4}$ s$^{-1}$, suggesting a solid-like state as also seen for other yield-stress fluids[24]. Despite complex temporal dynamics, the average steady-state shear rate shows very good agreement with the HB model for soft yield-stress materials: $\sigma = \sigma_y + \alpha\langle\dot{\gamma}\rangle^n$, with $n = 0.62$, $\sigma_y = 28$ Pa for $d = 10$ μm and $n = 0.81$, $\sigma_y = 29$ Pa for $d = 25$ μm. The value $n < 1$ indicates shear thinning, which is similar to the velocity-weakening nature of faults during earthquakes[33].

We show $\dot{\phi}$ as a function of time for different stress values ($\sigma$) as indicated in Fig. 1b for a sample thickness $d = 10$ μm. For very small stress values ($\sigma = 0.01$ Pa), no burst-like event is observed and $\dot{\phi}(t)$ fluctuates steadily in time. However, for $\sigma \geq 0.1$ Pa and beyond, a burst occurs with time span ~1000 s. Inside these bursts, the amplitude of $\dot{\phi}$ fluctuations (~2 mrad s$^{-1}$) (referred as "quakes" in this paper) is much larger than the average long-term angular velocity ($\langle\dot{\phi}_{\text{steady}}\rangle \sim 0.2 \times 10^{-4}$ mrad s$^{-1}$), revealing strong stress-induced organization in the sample. The center of the burst gets shifted toward longer times as the sample thickness ($d$) increases (Fig. 1c). We also find that in the time window of the burst in $\dot{\phi}$, there are large jumps in the cumulative strain $\gamma(t)$ ($\gamma(t) = \int_0^t \frac{R}{d}\dot{\phi}(t')dt'$) (Supplementary Fig. 4), signifying large-scale stress-induced reorganization in the system. However, after the burst, $\gamma(t)$ varies very slowly with time, implying that in this temporal region the system enters into a state with much larger shear modulus (about two orders higher in magnitude; Supplementary Fig. 1). For higher values of stress (>30 Pa), $\gamma(t)$ increases rapidly with time. Interestingly, for small applied stress values, the long-term strain can even reverse course as a result of the strong internal reorganization in the system. Such anomalous

flow behavior has been observed in the context of congested traffic flows, known as the Braess paradox[34]. One important feature of our systems is reversibility. After the burst-like event in $\dot{\phi}$, the system goes into a steady state over the remaining duration of the experiment (till 8000 s). However, if the sample is fluidized after the burst-like event by applying a stress larger than the yield stress ($\sigma > \sigma_y$) and then made to creep under a stress below the yielding ($\sigma < \sigma_y$), the system again shows a burst in the angular velocity (Supplementary Fig. 5). This indicates that fluidization erases the history of the reorganization of the system.

**Gutenberg–Richter law for angular velocity quakes**. We next focus on the statistical properties of these angular velocity quakes inside these burst events. We estimate the kinetic energy associated with each quake as $E = \dot{\phi}^2$ in units of mrad$^2$ s$^{-2}$. We find that the ensemble probability density function of kinetic energies $P(E)$ shows a robust power-law decay over more than three decades in $E$ (Fig. 2a): $P(E) \sim 1/E^\epsilon$, where $\epsilon = 1 + (b/1.5) = 1.6 \pm 0.1$; consistent with the Gutenberg–Richter law for the distribution of earthquake amplitudes[28], and is close to the value of $\epsilon$ obtained in other experiments[2,6,10]. The value of the Gutenberg–Richter exponent $b$ for our system ($=0.90 \pm 0.15$) is very close to the most accepted value for earthquakes ($b \sim 1$), but slightly greater than the reported values for other fracture experiments ($b \sim 0.60$)[8,9,11]. The exponent $b$ obtained via the maximum likelihood method[8] for different low-energy cutoffs ($E_{\text{cutoff}}$) is very stable for our system (inset Fig. 2a). $P(E)$ versus $E$ corresponding to different applied stress values and sample thicknesses are shown in Supplementary Fig. 6. We note that the relatively shorter range of power-law regime compared with other laboratory experiments cited earlier comes from the low-energy cutoff effects due to the background activity. The background activity is visible as the residual activity in the steady state (after 1500 s in Fig. 1b). This limitation in energy range is also seen in the probability density function with the mechanical energies of the continuously sheared granular matter[10], with the energy released in the numerical simulations of a system of bubbles[17], and in many other systems.

**Statistics of inter-occurrence waiting time**. The distributions of inter-occurrence waiting time of quakes for the three data sets corresponding to $d = 10$ μm (having energy cutoff $2.07 \times 10^{-2}$ in units of mrad$^2$ s$^{-2}$) follow a power-law behavior with exponent ~1.5 (Fig. 2b). We show the robustness of this power-law by taking different energy cutoff values for the $\sigma = 2$ Pa data (inset Fig. 2b).

**Omori law**. We now quantify the dynamics around the main shock of the burst. We perform a timescale analysis of the angular velocity $\dot{\phi}(t)$ using a wavelet transform, which is well suited for transient signals. The obtained wavelet coefficient at time $\tau$ and scale $a$ reads:

$$C_{\tau,a} = \frac{1}{\sqrt{a}} \int_{\text{time}} \dot{\phi}(t) \, \bar{\Psi}\left(\frac{t-\tau}{a}\right) \, dt, \qquad (1)$$

where $\bar{\Psi}$ stands for the conjugate of the mother wavelet, here chosen as the Morlet wavelet. The factor $\frac{1}{\sqrt{a}}$ ensures the normalization of $\bar{\Psi}\left(\frac{t-\tau}{a}\right)$. The Morlet wavelet has a dominant period $T_0$, so that a daughter wavelet $\bar{\Psi}\left(\frac{t}{a}\right)$ will thus feature a dominant period $T = aT_0$. In the remaining of this paper, we shall use the corresponding time period $T$ instead of the scale $a$, and study the wavelet coefficients $C_{\tau,T}$. Guided by Omori's law observed in

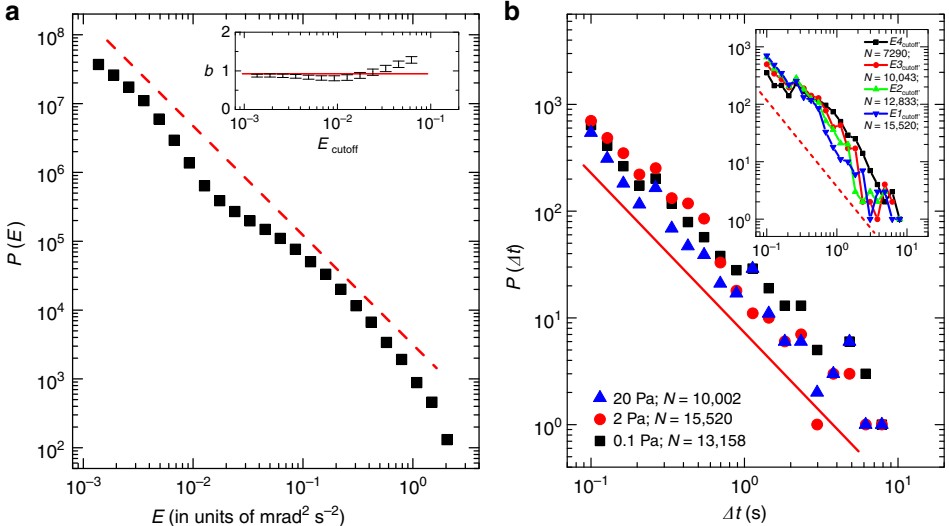

**Fig. 2 Statistics of kinetic energy of quakes (Gutenberg–Richter law) and inter-occurrence waiting time. a** Probability density of the kinetic energy ($P(E)$) versus kinetic energy ($E$) of quakes for $d = 10$ μm and $\sigma = 2$ Pa data (plotted in Fig. 1b) is shown. The red dashed line is drawn parallel to the linear fit with slope $\epsilon = 1.6 \pm 0.1$, indicating the Gutenberg–Richter exponent $b = 0.90 \pm 0.15$ (see the article for definition). Inset shows $b$ as a function of lower cutoff of energy $E_{cutoff}$, obtained via the maximum likelihood (ML) method, that is, considering only quakes with $E > E_{cutoff}$. Symbol heights are equal to the fitting errors in $b$. The horizontal red line represents $b = 0.9$ value. The results for other bursts are very similar (see Supplementary Fig. 6). **b** Distribution of waiting time ($\Delta t$, the time separation between successive quakes) for different $\sigma$ having $d = 10$ μm considering quakes with $E > 2.07 \times 10^{-2}$ in units of mrad$^2$ s$^{-2}$ is shown. Inset shows similar plot for 2 Pa data with different energy thresholds ($E1_{cutoff} = 2.07 \times 10^{-2}$, $E2_{cutoff} = 3.36 \times 10^{-2}$, $E3_{cutoff} = 5.46 \times 10^{-2}$, $E4_{cutoff} = 8.86 \times 10^{-2}$). Both, the red solid line and the red dashed line in the inset, show power-law decay with exponent 1.5. $N$ is the counted quakes of each analysis.

earthquake bursts, Eq. (1) predicts

$$\frac{|C(\tau_s, T)|}{\sqrt{T}} \sim \frac{1}{T^\alpha}, \qquad (2)$$

Where $\tau_s$ is the main-shock time. The power-law behavior of the wavelet coefficient near the singularity ($t = \tau_s$) reveals the finite-time singular power-law behavior of the angular velocity. To check for a possible asymmetry on the left versus right side of the singularity time $\tau_s$, we use the two distinct mother wavelets $\Psi_L$ and $\Psi_R$, where $\Psi_L$ is identical to the Morlet wavelet for $t \leq 0$ and zero for $t > 0$, while $\Psi_R$ is chosen to be zero for $t < 0$ and identical to the Morlet wavelet for $t \geq 0$. For all observed bursts, the wavelet coefficients are found to scale with the wavelet periods in a power-law fashion, almost over three decades of wavelet period with the value of the exponent $\alpha \sim 1$ as shown in Fig. 3a. We can also directly plot the rate of foreshocks and aftershocks as a function of time distance from the main shock to extract the Omori exponent (Fig. 3b). We find that Omori exponent of 1 fits the data very well for both fore-shock and after-shock regimes (Fig. 3b). For this system, the background amplitude during the foreshocks is greater than the one during aftershocks (so that the foreshocks $E_{cutoff}$ > aftershocks $E_{cutoff}$). After the background subtraction, the rate of foreshocks is lower than the rate of aftershocks for a given time distance from the main shock, similar to the rate asymmetry of real earthquakes. Only the data for $\sigma = 2$ Pa with $d = 10$ μm is plotted for clarity. For other stress values and sample thicknesses, the wavelet coefficients are plotted as a function of the wavelet period in Supplementary Fig. 7, where power-law behavior is obtained with $\alpha \sim 1$, similar to the statistics of the occurrence of aftershocks for the earthquakes[1,29,35]. While the inverse and direct Omori laws are recovered in their time dependence, there is an important difference with earthquakes: real foreshocks–aftershocks sequences are highly asymmetrical, the more so, the larger the main shock, with just a few events before the main shock (the number of foreshocks being independent of the size of the latter)

and a variable number of aftershocks. This asymmetry reflects the productivity law, stating that the number of triggered events grows exponentially with the main-shock magnitude. This productivity law results from the extended nature of the rupture and the interactions within complex fault networks. It is thus naturally expected that below some magnitude threshold (depending on the specific productivity parameters) earthquakes feature less aftershocks than foreshocks.

**Quantitative in situ rheomicroscopy of domain reorganization.** Using the birefringent properties of the nematic phase of the surfactant, we capture the dynamics of nematic domains under stress by a home-built POM setup (see the Methods section). Even in the huge burst, the largest angular velocity quake corresponds to a displacement of the top plate by ~1 μm, which is well below the resolution of the optical imaging system. However, we can probe the reorganization in the system by tracking the changes in the domain structure. We use transparent circular glass plates as top and bottom plates of the rheometer with a gap between them ~150 μm. We capture images at a frame rate of 0.1 Hz with a CCD camera (Methods). The details of the setup are described in Supplementary Fig. 8. In Fig. 4b–e, we show the evolution of the intensity pattern as a function of time for an applied stress $\sigma = 2$ Pa. Over time, these structures show a coarsening effect due to the orientational ordering of domains (Supplementary Movie 1). We find that when the applied stress is zero, no such evolution of intensity pattern is observed (Supplementary Movie 2), further confirming that such ordering in domains is due to the internal reorganization of the system under applied shear stress. Structure coarsening is quantified by calculating the spatial auto-correlation of intensity ($A(\mathbf{k}, t)$) in the wave vector ($\mathbf{k}$) domain at different times $t$. For $t < 2000$ s, $A(\mathbf{k}, t)$ decays with increasing $\mathbf{k}$ and $t$ (Fig. 4f). After 2000 s, it does not evolve much with time. To bring out the largest length scale of the structures, we plot $\mathbf{k}_{max}^{-1}$ as a function of time, where $A(\mathbf{k}_{max}, t) = 0.2$ (Fig. 4g). It can be clearly seen that, after ~2300 s, the

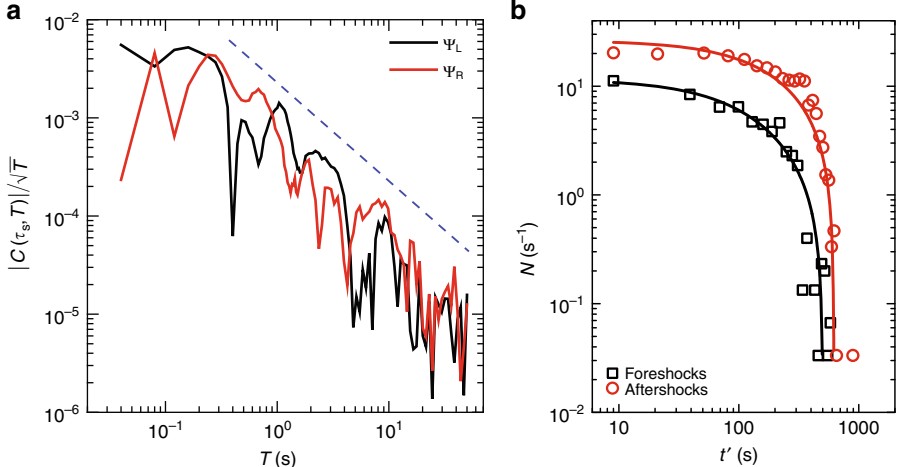

**Fig. 3 Statistics of rate of quakes: Omori law. a** Log–log plot of $|C(\tau_s, T)|$, the wavelet coefficient (normalized by $\sqrt{T}$) before (using $\Psi_L$ as the analyzing wavelet; points on the left portion of the burst) and after (using $\Psi_R$ as the analyzing wavelet; points on the right portion of the burst) the main shock, as a function of the wavelet period $T$ of the angular velocity for $\sigma = 2$ Pa, $d = 10$ μm (Fig. 1b). The slope suggests a singularity (the blue dashed line indicates a power-law decay with exponent $\alpha = 1$). The results for other bursts are shown in Supplementary Fig. 7. **b** The rate of quakes ($N$) before and after the main shock ($\tau_s = 594$ s; having foreshocks $E_{cutoff} = 2.31 \times 10^{-1}$, aftershocks $E_{cutoff} = 2.57 \times 10^{-2}$) of the same data is plotted versus time distance from the main shock ($t' = |t - \tau_s|$). The counting time window $\Delta t = 30$ s is taken to get good statistics in $N$. Fitted curves have the form $N = N_0 + A/(c + t')^p$ with $p = 1$ and $c$ has values $185 \pm 43$, $389 \pm 10$, respectively.

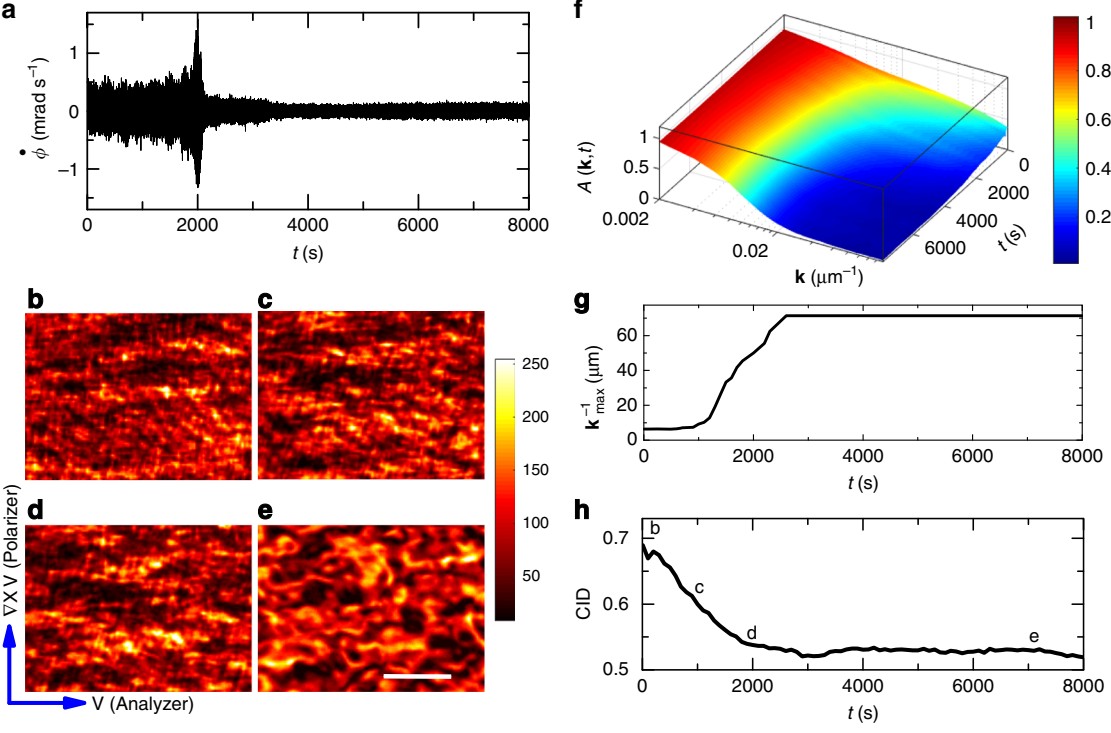

**Fig. 4 In situ polarized optical microscopy of domain reorganization in CTAT nematics. a** $\dot{\phi}$ versus $t$, measured during the rheo-POM experiment with 39 wt% CTAT + water sample at 30 °C ($d = 150$ μm, $\sigma = 2$ Pa). POM images obtained at different times **b** 10 s, **c** 1000 s, **d** 2000 s, **e** 7000 s show coarsening of domains, polarization axis, and flow direction are indicated, colors represent intensity level, scale bar has length 50 μm. **f** Auto-correlation ($A(\mathbf{k}, t)$) as a function of the wave vector ($\mathbf{k}$) for different times, colors represent different $A(\mathbf{k}, t)$ values. **g** We plot the largest length scale $\mathbf{k}_{max}^{-1}$ versus $t$; where $A(\mathbf{k}_{max}, t) = 0.2$. **h** Computable information density (CID) is plotted with time; b, c, d, e markings are for the images shown above. $A(\mathbf{k}, t)$ and CID for no-shear condition ($\sigma = 0$ Pa) are plotted in Supplementary Fig. 9.

structural reorganization is minimal. Another measure of this is the recently discovered computable information density (CID), which is proportional to the losslessly compressed image size[36]. CID is defined as CID $\equiv L(t)/L$, where $L(t)$ is the size of the compressed image and $L$ is the original size. All our original uncompressed "BMP" images are of the same size ($L = 602$ KB).

We plot CID as a function of time, using $L(t)$ the size of the images compressed to "PNG" format (Fig. 4h). We find that the image sizes drop until the system reaches a steady state after the burst. This indicates that the structures are getting bigger in size until the main shock appears. Since the POM captures the orientational order in the system, a larger domain structure

indicates a larger range of local orientational ordering in the system. This observation signifies that the domains form a jammed structure with the average nematic domain size $\sim 70$ μm to counter the applied stress against any further rearrangements in the sample.

**Flow of colloidal glass of laponite nanodiscs.** To ascertain that the statistical properties of $\dot{\phi}$-quakes described so far are generic to other yield-stress solids, we have also studied a repulsive Wigner glass formed by 5 wt% laponite clay suspension. Laponite clay particles are disk-shaped having a diameter $\sim 20$ nm and thickness 2 nm[37]. The particles have a weak positive charge along the rim but a net negative surface charge that makes the interparticle interactions long ranged[32]. Even without the application of any shear stress, these systems show considerable aging effects where viscosity can increase by orders of magnitude with waiting time[38]. An important issue while doing rheology with aging samples is to get a reproducible initial condition, because the process of sample transfer and loading in the rheometer involves shearing of the sample. We use the following protocol to get a reproducible initial condition[27]: after loading the sample on the rheometer plate, we apply a small constant shear rate of 0.1 s$^{-1}$ for 1000 s. Then the sample is rejuvenated by applying a large stress of 50 Pa for 500 s. As soon as this process is over, we apply the desired stress ($\sigma$) for which we want to monitor the time dependence of the angular velocity ($\dot{\phi}$). We summarize our experimental data of 5 wt% laponite clay suspension with $\sigma = 15$ Pa and $d = 40$ μm and 10 μm in Fig. 5. The angular velocity $\dot{\phi}$ as a function of time is shown in Fig. 5a, where we again see burst-like events. In this case also, the quakes follow the Gutenberg–Richter law: $P(E) \sim 1/E^{\epsilon}$; $\epsilon \sim 1.5$ (Fig. 5b) with

Gutenberg–Richter exponent $b \sim 0.75$. The waiting time distributions show exponent $\sim 1$ (Fig. 5c). The wavelet coefficients are again found to scale with the wavelet periods in a power-law fashion (Eq. (2)) over three decades of wavelet period with the exponent $\alpha \sim 1$ as shown in Fig. 5d. Samples with different laponite concentrations show similar bursts, but with different quake amplitudes (Supplementary Fig. 12). For this system also, we can simply erase the memory by applying the pre-shear protocol (Supplementary Fig. 13). The laponite system is much more complicated due to the intrinsic aging of the sample at rest (see Supplementary Fig. 14 for experiments with different aging times). We found that the Gutenberg–Richter exponent $b = 0.75 \pm 0.05$ (summarizing all our results for Laponite) is very stable with respect to the different conditions imposed on the laponite sample here. However, after the primary burst, there are several secondary ones, unlike in the surfactant system. Such behavior probably arises from the complex structural evolution resulting from the competition between intrinsic and shear-induced aging effects. Secondary bursts appearing in $d = 10$ μm data are analyzed, and they are also showing power-law behaviors (Supplementary Figs. 10, 11). We would like to point out that even for this system, when the aging is sufficient to drive the system into a glassy state, the stress-induced yielding shows very similar behavior like the nematic phase of CTAT (see Fig. 1 of ref. [27]). Laponite colloidal glasses do not show any birefringence property over the concentration range we have studied, and hence no imaging could be done.

## Discussion

In our experiments, the observation of both positive and negative angular velocity fluctuations as a result of a unidirectional drive (constant applied stress) can be rationalized by the physical

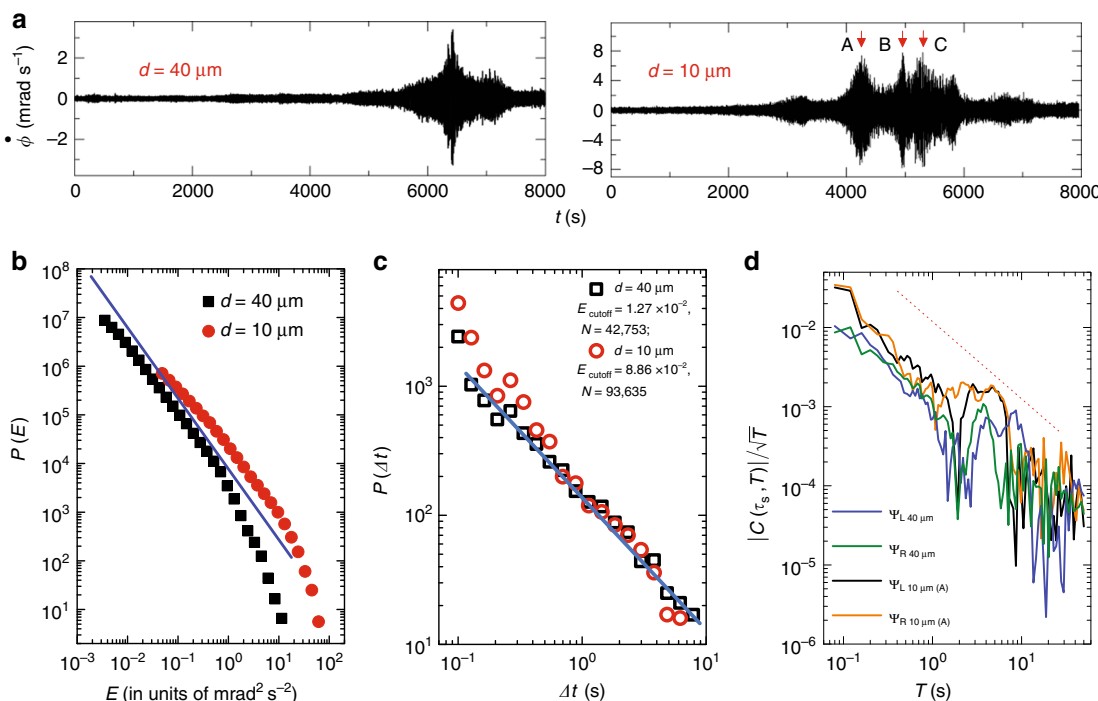

**Fig. 5 Flow behavior of colloidal glass of Laponite nanodiscs and statistical analysis of the quakes. a** Data from a completely different sample with 5 wt% concentration of Laponite clay for two different $d$ with $\sigma = 15$ Pa (at 26 °C) are shown. With this sample, $\dot{\phi}$ sometime shows multiple bursts (arrows with marks A, B, C in $d = 10$ μm data). **b** Log–log plot of $P(E)$ versus $E$. Blue line has slope $-1.5$. **c** Waiting time distribution plot. Straight line has slope $-1$. We have considered full time window of the data for $P(E)$ and $P(\Delta t)$ plots. **d** $|C(\tau_s, T)|/\sqrt{T}$ versus $T$ plot for both $d$'s; wavelet transformation for $d = 10$ μm data set was done considering the burst near A ($\sim 4300$ s). Red dashed line has slope $-1$. The results for the other bursts near B and C are shown in Supplementary Figs. 10 and 11.

picture proposed by Nandi et al.[39], based on a stochastic attachment and detachment dynamics of surfactant micelles/laponite particles with the rotating top plate of the rheometer. As the sample deforms, micellar domains/particle clusters create transient bridges between the two plates that form and break intermittently, leading to a very jerky motion (strain) of the drive similar to stick–slip dynamics[10,40], with the angular velocity showing large fluctuations (quakes) of both positive and negative signs. Within this physical picture, the average strain-strengthening behavior after the burst can be associated with jamming of such bridges that leads to an increase of the average long-time viscosity. The bridges constitute force paths, similar to the force chains in proposed analogy between granular media, tectonic deformations, and spinglasses[41,42] and to the stress field in model of fault network in the Earth crust[43,44]. The coarsening of nematic domains in the micellar system with time under an applied stress likely indicates the growth of such jammed regions in the system whose reorganization leads to a stronger solid-like state.

While not being perfect analogs, our experimental systems can be put in correspondence with the seismogenic crust via dimensional scaling of lengths, times, and mechanical properties, such as strain and stress. For this, we introduce the scaling factors $L_{sf} = L/L^*$ and $T_{sf} = T/T^*$ relating the length $L^*$ and timescales $T^*$ of the model to those, $L$ and $T$, in the Earth crust. The spatial range over which tectonic shear stresses occur in the Earth crust due to tectonic motions is of the order of 100 km. The width of the gap over which the shear stress is applied in the experiment is ~10 μm. Thus $L_{sf} \sim 10^{10}$. The typical duration of earthquake clusters (including foreshocks–mainshock–aftershocks) is of the order of months to years. The typical time span of a burst in the laboratory experiment is about 1000 s, thus we have $T_{sf} \sim 10^5$. From these two scaling ratios $L_{sf} \sim 10^{10}$ and $T_{sf} \sim 10^5$, we proceed to derive the dimensional scaling laws of all other relevant physical quantities, allowing to map our laboratory experiment onto the Earth crust.

An earthquake in the crust is associated with the driving shear stress overpassing the frictional force along faults. The frictional force is equal to the normal stress, controlled by the lithospheric pressure, times the friction coefficient, in the range 0.1–0.6. Hence, the scale of the stresses at which fault sliding is triggered is governed by $\rho g h$, where $\rho$ is the average density of the Earth crust, $g$ is the acceleration of gravity, and $h$ is the width of the seismogenic crust. As $\rho$ is about three times that of our suspensions, and $g$ is the same, this predicts that the shear modulus in our experiments should be about $1/L_{sf}$ that of the Earth crust (roughly $10^{10}$ to $10^{12}$ Pa), i.e., 1–100 Pa in the experiments, which is of the same order as in our laboratory observations. The typical rupture sliding velocity during an earthquake is $\sim 1 \, m \, s^{-1}$, which should be $L_{sf}/T_{sf} \sim 10^5$ times the velocity of local peaks of the burst in our experiments. Indeed, the typical velocity of the shear motion at a local peak is $\sim R\dot{\phi} \sim 1 \, cm \times 1 \, mrad \, s^{-1} = 10^{-5} \, m \, s^{-1}$. Lastly, since strain is dimensionless, the strain rate of $\sim 0.1 \, s^{-1}$ in our sample at a peak during a burst should correspond to $T_{sf} = 10^5$ times the strain rate during an earthquake, which is typically 1 m of slip over a fault length of 100 km occurring in 10 s of slip, which yields the strain rate $\sim 10^{-6}$, which is in perfect agreement with the prediction from the scaling laws.

We have shown a good semi-quantitative description of our experimental data with Gutenberg–Richter law, Omori scaling law, and the power-law distribution of inter-occurrence waiting time over a wide range of stress values and system sizes. The robustness of the scaling exponents observed in our experiments and their similarity with the values obtained for other systems having orders of magnitude different moduli suggest an underlying universal mechanism of stress-induced reorganization in different materials via the general process of avalanches triggered by threshold dynamics[45]. Our experiments on yield-stress materials may help to correlate such scaling laws (and especially deviations from them) directly to the microscopic deformations in the system.

The reversal of the low-frequency angular velocity is reminiscent of an analog in congested traffic flows, known as the Braess paradox[34]: when a new link is added to a random network (a new cluster or bridge forms), the capacity can be augmented or decreased with equal probability[46]. The relevance of this mechanism for mechanical networks has been exemplified by Cohen and Horowitz[47]. In addition to the quantitative analogies with the inverse and direct Omori laws, one can observe that the angular velocity exhibit seismogram-like structure having many quakes in it. This parallels the succession of repetitive seismic bursts, accelerating with time, and increased low-frequency seismic noise, which was observed for one of the best-recorded large earthquakes to date, the 1999 7.6 Mw Izmit (Turkey) earthquake[48].

It is tempting to interpret the log-periodic variation of wavelet coefficients seen near the singularity (Fig. 3). Such behavior has been observed in case of earthquakes[35], material failures[49], finance, and population dynamics[50], indicating a discrete-scale invariance property of the system as opposed to continuous-scale invariance given by ordinary power laws. However, such study is outside the scope of this paper. We hope that our work will motivate further experimental and theoretical studies in such subtle but fascinating stress-induced reorganizations in a wide range of materials.

## Methods

**Procedure of rheology experiments.** All the rheology experiments are carried out using a MCR 102 stress-controlled rheometer (Anton Paar, Austria) fitted with PT temperature controller using 25-mm steel parallel-plate (P-P) geometry. A humidity chamber is used to minimize the evaporation of water from the sample during the rheology experiments. The bottom plate of the instrument is fixed, and the top plate is connected to the torque transducer as shown in Supplementary Fig. 2. The minimum angular resolution of this instrument is 0.01 μrad. The details of the experimental setup are same as in ref. [26]. Due care is taken to cutoff all external sources of vibrations using vibration isolation arrangements. For Rheo-POM (rheology-POM) measurement, steel plates are replaced by glass plates with the polarizer axis along the vorticity direction (Supplementary Fig. 8). The images were grabbed at a 5 mm interior from the edge of the top plate, with an eight-bit color CCD camera (Lumenera, 0.75C, 640 × 480 pixels) fitted with a microscope (objective: M Plan Apo 5X, N.A. 0.14) and analyzed in ImageJ.

**Preparation of CTAT-water system.** Cetyltrimethylammonium Tosylate (CTAT)-water samples are prepared by dissolving known amount of CTAT (Sigma Aldrich) in ultrapure deionized double-distilled water (Millipore) at pH 10 in 10-ml vials, and are kept sealed for equilibration for two weeks at 60 °C.

**Preparation of laponite–water system.** The laponite powder as procured from Southern Clay Products Inc. is mixed with deionized water for 10 min with a magnetic stirrer until a clear solution is obtained. The solution is then passed through a 0.45 μm pore size filter to obtain the final sample. During the filtration process, the sample is under very high stress/strain rate, and hence we define the zero of the aging time for the sample after the filtration process is over.

## Data availability

All the data that support the findings of this study are available from the corresponding author upon reasonable request.

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

## Acknowledgements

A.K.S. thanks Department of Science and Technology (DST), India for support under Year of Science Fellowship. S.M. acknowledges Science and Engineering Research Board (SERB, under DST) for support through a Ramanujan Fellowship. P.K.B. thanks University Grants Commission (UGC), India for the Senior Research Fellowship. We thank Prof. Rajesh Ganapathy for critical reading of the paper and Prof. V.A. Raghunathan, Dr. Aditya Sood for fruitful discussions.

## Author contributions

P.K.B., S.M., G.O., D.S., and A.K.S. designed the research; P.K.B. and S.M. did the experiments. P.K.B., S.M., G.O., D.S., and A.K.S. contributed to the analysis and preparation of the paper.

## Competing interests

The authors declare no competing financial or nonfinancial interests.
