## [Peer Review File · Nature Communications]

Reviewers' comments:

Reviewer #1 (Remarks to the Author):

This is a very nice paper. The experimental conditions are non-trivial and the authors did a good job in obtaining excellent raw data. The methods to use the 'first derivative square method' to identify jerks works excellently in this case. The statistical analysis is as good as possible for such data sets. The analysis has been done before in other systems by Sornette and was shown to be appropriate. I like in particular the optical methods which illustrate the avalanche mechanisms very well.

Some minor points:

the power law regime is surprisingly small and is even smaller in the second experiment. Here I suspect low energy cut-off effects to be dominant. Some more comments would be helpful.

The epsilon value of 1.5 should be given explicitly (in courtesy for cursory readers, and not just b) but with error bars. I would guess that ± 0.15 would be fair which covers most of similar results. In particular, it covers the collapse data in Intermittent flow under constant forcing by Salje et al. (APPLIED PHYSICS LETTERS 112 : 054101, 2018) but not the creep data where the exponent is clearly higher. The question is then: is the strain rate slow enough in the current experiment? The idea that creep may play a role is evoked. I would have liked some more comments.

Finally: the waiting time analysis was not undertaken. Why? Where the data not good enough?

These points are minor compared with the success of the paper. We have really big problems in understanding Earth Quakes where, in recent cases, no Omori quakes were found and where we are still not very successful in finding precursor signals. This paper has shifted the focus on 'soft' materials. It should be published.

Reviewer #2 (Remarks to the Author):

The article of Bera et al analyses the mechanical signals linked to structural reorganisations of two yield solids below the yield stress. Their main original claim is the following: "For the first time we find that, for small sample thicknesses (the gap d between shearing plates of a few tens of microns), the angular velocity fluctuations show burst-like events (Fig. 1b,c) persisting over about thousands of seconds (comparable to seismic foreshock to aftershock time scale)"; and also the fact of showing a burst statistics following Gutenberg-Richter-like (G-R) relations.

These fluctuations, and plausible explanations, have been already reported by the authors in previous papers (ref. 15-16, 27), thus the main originality and relevance of this manuscript are the seismic-like behaviour of the dynamics. However, I find the main motivation (Earth crust should be considered as an effective very weak soft medium) misleading, and I do not see either the similarities between the reported results and earthquake dynamics, besides of the fact of having a power law distribution of events' energies (G-R), which is an attribute of many different natural systems. Please, find more detailed comments and questions below.

1- The main motivation of using very soft materials as analogue experiments of earthquakes: "At these large scales, the elastic moduli become small compared with gravity and the Earth crust should be considered as an effective very weak soft medium."

- Earthquake analogue experiments commonly use actual rocks, which are tested under the pressure found within a real fault at different depths (Passetlègue 2013, Schubnel 2013). Granular fault experiments with different materials, from quartz (Scuderi 2016) to relatively soft ($Y=100$ MPa) 3D-polymers (ref. 5), also mimics similar degrees of deformation (Applied stress/ Young modulus ($Y \sim 10^{(-4)}$))... so the authors need to better explain their "motivation" phrase, which (if true) carries a very relevant statement.

- The fact that very soft materials do not produce acoustic emissions is also a great disadvantage for using them as analogue experiments of earthquakes (i.e., as a system that can bring a relevant input into the understanding of the actual phenomenon).

- The application of a controlled shear stress to the sample contrast also with the real nature of a tectonic fault, which is driven at a relative constant shear velocity.

- Notice also the existence of analogue experiments with gels (Yamaguchi 2011).

- FX Passelègue, A Schubnel, S Nielsen, HS Bhat, R Madariaga, From sub-Rayleigh to supershear ruptures during stick-slip experiments on crustal rocks, *Science* 340 (6137), 1208 (2013).

- A Schubnel, F Brunet, N Hilairet, J Gasc, Y Wang, HW Green, Deep-focus earthquake analogs recorded at high pressure and temperature in the laboratory, *Science* 341 (6152), 1377 (2013).

- MM Scuderi, C Marone, E Tinti, G Di Stefano, C Collettini, Precursory changes in seismic velocity for the spectrum of earthquake failure modes, *Nature geoscience* 9 (9), 695 (2016).

- T Yamaguchi, M Morishita, M Doi, T Hori, H Sakaguchi, JP Ampuero, Gutenberg- Richter's law in sliding friction of gels, *Journal of Geophysical Research: Solid Earth* 116 (B12), (2011).

2- Concerning one burst: "However, for $\sigma > 0.1$ Pa and beyond, a burst occurs at ~ 500 s that persists for ~ 1000 s." (on figure 1b,c)

- It is not clearly written in the manuscript what is the equivalent of this burst in the case of real earthquakes. It took me a second reading to realise (by reading the main claims of the paper in the introduction) that one burst corresponds to a sequence foreshock-aftershocks. Am I right? If so, it would be nice to better describe the dynamics: e.g. how many burst, and how different are they during one typical experiment?

- Figure 2 indicates a G-R relation in the "statistical properties of bursts". To avoid the confusion, I think that it is important (if the authors decide to continue on exploiting the similitude with earthquakes) to give a name to their equivalent to "one earthquake", which may be one spike within a burst. During my first reading I was considering that one burst was one earthquake and I was wondering how many events of 1000 s were analysed in the statistics?

3- If my analysis so far is correct, there is one mainshock in each burst, defining foreshocks to the left and aftershocks to the right of it. It would be interesting to analyse the statistics of mainshocks to see if they follow a G-R law, also it is important to indicate the origin of the spikes analysed in figure 2: From how many bursts? How different are those bursts? Are they from different experiments or they all are from the same experiment?

4- The exponent Beta is not constant, but varies from around 0.6 to more than 1.0, which is huge in terms of earthquakes (see Supp. Mat in ref. 5). Maybe there is some interesting physics behind. See for example (Barés 2018).

-J Barés, A Dubois, L Hattali, D Dalmas, D Bonamy, Aftershock sequences and seismic-like organization of acoustic events produced by a single propagating crack, *Nature communications* 9 (1), 1253 (2018).

5- Why the need of wavelets analysis to introduce an Omori law, which is quite easy to analyse in its original form? Presenting the original way (if possible) must be a stronger indication of the similarity to actual earthquakes.

Reviewing on the dynamics: The foreshocks/aftershocks sequences are not similar to the ones presented in real earthquakes. In nature there is a very important asymmetry between foreshock and aftershocks activities (very few foreshocks against a very large number of aftershocks). In this

manuscript we find the opposite asymmetry with a large and progressive activity in the foreshock zone. The only similarity is the G-R law, which is an attribute of many different natural systems, so I do not consider that this system presents an earthquake-like dynamics that make it of interest for a community working on earthquake physics.

Reviewer #3 (Remarks to the Author):

The aim of the work is to mimic earthquake-like statistics (Gutenberg-Richter energy-release distribution, Omori aftershock time-decay) in laboratory experiments. The authors refer to earlier experiments with such statistics on hard materials (rock, ceramic, wood) where earthquake-like statistics has been analyzed with acoustic emission (AE). They correctly argue that true comparison requires soft model systems, for which AE cannot be used. Their choice is two soft yield-stress fluids, which they analyze below the yield stress with shear rheology and polarized optical microscopy. The observed statistics of energy bursts under shear indeed corresponds with the Gutenberg-Richter (GR) law, with exponent $b = 0.75$; the temporal behavior nicely mimicks Omori's law of aftershocks, with creep exponent a around unity. In both cases the scaling exponents are close to what is seen for earthquakes (b and a both around unity). In the interpretation the authors propose a picture of intermittent formation and breakup of bridging clusters of the colloidal particles across the rheometer gap. In general the goals are very relevant and this work seems well worth publishing.

There are however a number of objections or questions. First of all the authors state that "to our knowledge there is no similar study on soft and disordered materials that show solid-like yield stress". There is earlier work in soft systems that have been measured by other ways than AE, and that is very relevant in the context. Although these soft systems concern not always jamming yield-stress fluids they too show the same earthquake-like statistics, e.g.:

- Sprakel et al., Phys. Rev. E 79, 056306 (2009) already report intermittent dynamics with power-law stress-drop statistics in experiments of soft transient networks under shear; they propose a similar picture of fracturing and restoring network bonds. In this case the system is fluid at zero shear and has a soft stress peak in the flow curve; it causes stress fluctuations between a high- and low-viscosity state, which shows that jamming bridges across the full gap and true stick-slip are not necessary to give earthquake-like behavior: the system may still be a high-viscosity fluid at very long (e.g. geological) timescales.

- Zargar et al. have reported Gutenberg-like statistics in soft colloidal glasses: Phys. Rev. Lett. 110, 258301 (2013)

- Yamaguchi et al., J. Geophys. Research 116, B12306 (2011) publish GR statistics in stick-slip sliding friction of a hard block across a soft gel plate.

- Siebenburger et. al., Phys. Rev. Lett. 108, 255701 (2012) have measured logarithmic creep ($a = 1$) in the long-time regime for sheared colloidal glasses well below the yield stress.

In addition to these experimental studies, a Lattice-Boltzmann simulation by Benzi et al., Geophys. J. Int. 207, 1667 (2016) reveals the same GR and Omori statistics as above for plastic events in a soft glass.

These earlier papers need to be discussed in detail in order to assess what is really novel here, and what is not. Provided that also questions on other aspects of the present manuscript can be satisfactorily answered, the work may still be considered a valuable extension of existing studies, with

new model systems and novel and in some respects more detailed ways of measurement and analysis, deserving publication in a very good journal.

I also have the following questions:

- The authors suggest that their experiments, in contrast to those on the hard systems, favorably compare with earthquakes a.o. because of the similar absolute value of the sampling rate, measured in Hz. Should a true correspondence between strongly differing systems such as geological faults and colloids not rather be based on comparison of dimensionless timescales and stresses, using characteristic times and energies of each system in the reduction? Can the authors provide such a scaling?
- The Laponite system is introduced to ascertain the conclusions on the nematic system. But the former is aging at rest. To what extent do the results change with changing preparation protocol? To what extent may the timescale of aging interfere with the timescale of bursts? To what extent can the aging system actually be compared with the Herschel-Bulkley nematic system and can the same interpretive picture be used?
- Can the authors give more quantitative information on the size of the dynamically coherent domains compared to the gap size
- Dinkgreve et al. *Journal of non-Newtonian fluid mechanics* 238, 233-241 (2016) perform similar creep experiments on a similar soft system, but do not see any intermittency. Do the authors have any idea what is specific to their systems, and what is generic?

Reviewers' comments:

We thank the reviewers for their detailed and helpful remarks. We have modified the manuscript taking these into account.

Reviewer #1 (Remarks to the Author):

This is a very nice paper. The experimental conditions are non-trivial and the authors did a good job in obtaining excellent raw data. The methods to use the 'first derivative square method' to identify jerks works excellently in this case. The statistical analysis is as good as possible for such data sets. The analysis has been done before in other systems by Sornette and was shown to be appropriate. I like in particular the optical methods which illustrate the avalanche mechanisms very well.

>> We thank the reviewer for finding our work interesting and substantial.

Some minor points:

(i) the power law regime is surprisingly small and is even smaller in the second experiment. Here I suspect low energy cut-off effects to be dominant. Some more comments would be helpful.

(ii) The epsilon value of 1.5 should be given explicitly (in courtesy for cursory readers, and not just b) but with error bars. I would guess that ± 0.15 would be fair which covers most of similar results. In particular, it covers the collapse data in Intermittent flow under constant forcing by Salje et al. (APPLIED PHYSICS LETTERS 112 : 054101, 2018) but not the creep data where the exponent is clearly higher. The question is then: is the strain rate slow enough in the current experiment? The idea that creep may play a role is evoked. I would have liked some more comments.

>> (i) The relatively small range of power law regime in our case indeed comes from the low-energy cut-off effects as pointed out by the referee. As shown in Fig.1, Fig.4 and Fig.5, we find that a burst like event relaxes to a steady-state region ($t > 1500$ s for Fig.1b). However, in the steady state substantial residual fluctuations (background activity) are still present which determines the lowest energy cut-off values to be considered and restrict the range for power-law regime. This is also true for fluctuations before the burst appears (shown in Fig.1c, Fig.4a). This limitation is also seen in the probability density function with mechanical energies of continuously sheared granular matter (Lherminier S, et al., PRL, 2019), in the probability distribution of the energy released in the numerical simulation with system of bubbles (Benzi R, et al., Geophys. J. Int., 2016), and in many other systems.

We have added a few lines to highlight this important point in the revised manuscript.

(ii) Following the referee's suggestion, we now explicitly indicate the value of the fitting power law parameter ϵ with error (1.6 ± 0.1) (Fig.2a). The corresponding error in b is also indicated (0.90 ± 0.15).

The average shear rate in our experiments (Fig.1b, 1c for CTAT) is $\sim 10^{-4} \text{ s}^{-1}$ as shown in Fig.1a. This is also the case for laponite (Fig.5a). The value of the average shear rate is indeed very low, and our experiments correspond to the creep flow region. The value of exponent $\epsilon = 1.6$ for CTAT (Fig.2a) is close to the value of ϵ obtained using the acoustic emission data from natural sandstone during creep (Salje, et al., APL, 2018). We further note that such exponent remains robust over a range of energy cut-offs (inset, Fig.2a) and applied stress values (Fig. S6).

This discussion has now been added in the revised manuscript with reference to the paper by Salje, et al., APL, 2018.

Finally: the waiting time analysis was not undertaken. Why? Where the data not good enough?

These points are minor compared with the success of the paper. We have really big problems in understanding Earth Quakes where, in recent cases, no Omori quakes were found and were we are still not very successful in finding precursor signals. This paper has shifted the focus on 'soft' materials. It should be published.

>> Guided by the reviewer's comments, we have now analysed the waiting time distribution with our data obtained with the nematic system (CTAT) as well as with the Wigner glass (laponite). The results of these new analysis are now included in Fig.2b & Fig.5c. Our analysis clearly shows that the waiting time distributions follow a robust power-law behaviour with exponents ~ 1.5 for CTAT (Fig.2b) and ~ 1 for laponite (Fig.5c). The value of the exponent of the waiting time distribution for laponite is close to the value observed for actual earthquake data from (1984 –2000) California region $20^{\circ}\text{N} - 45^{\circ}\text{N}$ latitude and $100^{\circ}\text{W} - 125^{\circ}\text{W}$ (Bak P, et al. PRL, 2002). We also note that a larger value of this exponent for CTAT system is close to the value (1.34) for distribution of inter-occurrence waiting time of acoustic emission due to crack propagation in an artificial rock formed by sintered monodisperse polystyrene beads (Bares J. et al., Nat. Comm., 2018). This discussion has now been added to the revised manuscript.

Reviewer #2 (Remarks to the Author):

The article of Bera et al analyses the mechanical signals linked to structural reorganisations of two yield solids below the yield stress. Their main original claim is the following: "For the first time we find that, for small sample thicknesses (the gap d between shearing plates of a few tens of microns), the angular velocity fluctuations show burst-like events (Fig. 1b,c) persisting over about thousands of seconds (comparable to seismic foreshock to aftershock time scale)"; and also the fact of showing a burst statistics following Gutenberg-Richter-like (G-R) relations.

These fluctuations, and plausible explanations, have been already reported by the authors in previous papers (ref. 15-16, 27), thus the main originality and relevance of this manuscript are the seismic-like behaviour of the dynamics. However, I find the main motivation (Earth crust should be considered as an effective very weak soft medium) misleading, and I do not see either the similitudes between the reported results and earthquake dynamics, besides of the fact of having a power law distribution of events' energies (G-R), which is an attribute of many different natural systems. Please, find more detailed comments and questions below.

1- The main motivation of using very soft materials as analogue experiments of earthquakes: "At these large scales, the elastic moduli become small compared with gravity and the Earth crust should be considered as an effective very weak soft medium."

- Earthquake analogue experiments commonly use actual rocks, which are tested under the pressure found within a real fault at different depths (Passelègue 2013, Schubnel 2013). Granular fault experiments with different materials, from quartz (Scuderi 2016) to relatively soft ($Y=100$ MPa) 3D-polymers (ref. 5), also mimics similar degrees of deformation (Applied stress/ Young modulus ($Y \sim 10^{-4}$))... so the authors need to better explain their "motivation" phrase, which (if true) carries a very relevant statement.

>> Modelling large scale phenomena such as tectonic events in the laboratory requires to scale properly both the loading stresses and the physical properties of the material. In order to establish the correspondence between the laboratory model and geological scales, we need to introduce the scaling factors $L_{sf} = L/L^*$ and $T_{sf} = T/T^*$ relating the length L^* and time scales T^* of the model to those, L and T , in the Earth crust. The spatial range over which tectonic shear stresses occur in the Earth crust due to tectonic motions is of the order of 100 km. The width of the gap over which the shear stress is applied in the experiment is ~ 10 micron. Thus $L_{sf} \sim 10^{10}$. The typical duration of earthquake clusters (including foreshocks-mainshock-aftershocks) is of the order of months to years. The typical time span of a burst in the laboratory experiment is about 1000 s, thus we have $T_{sf} \sim 10^5$. From these two scaling ratios $L_{sf} \sim 10^{10}$ and $T_{sf} \sim 10^5$, we proceed to derive the dimensional scaling laws of all other relevant physical quantities, demonstrating that our laboratory experiment is a good analog to the Earth crust.

An earthquake in the crust is associated with the driving shear stress overpassing the frictional force along faults. The frictional force is equal to the normal stress, controlled by the lithospheric pressure, times the friction coefficient, in the range 0.1-0.6. Hence, the scale of the stresses at which fault sliding is triggered is governed by ρgh , where ρ is the average density of the Earth crust, g is the acceleration of gravity and h is the width of the seismogenic crust. As ρ is about 3 times that of our suspensions, and g is the same, this predicts that the shear modulus in our experiments should be about $1/L_{sf}$ that of the Earth crust (roughly 10^{10} to 10^{12} Pa), i.e. 1 to 100 Pa in the experiments, which is of the same order as in our laboratory

observations. The typical rupture sliding velocity during an earthquake is ~ 1 m/s, which should be $L_{sf} / T_{sf} \sim 10^5$ times the velocity of local peaks of the burst in our experiments. Indeed, the typical velocity of the shear motion at a local peak is $\sim R d\phi/dt \sim 1 \text{ cm} \times 1 \text{ mrad/s} = 10^{-5}$ m/s. Lastly, since strain is dimensionless, the strain rate of $\sim 0.1 \text{ s}^{-1}$ in our sample at a peak during a burst should correspond to $T_{sf} = 10^5$ times the strain rate during an earthquake, which is typically 1 m of slip over a fault length of 100 km occurring in 10 s of slip, which yields the strain rate $\sim 10^{-6}$, which is in perfect agreement with the prediction from the scaling laws. We have added this discussion in the revised manuscript.

- The fact that very soft materials do not produce acoustic emissions is also a great disadvantage for using them as analogue experiments of earthquakes (i.e., as a system that can bring a relevant input into the understanding of the actual phenomenon).

>> As we have shown above, the scaling relations between our laboratory model and real earthquakes in the Earth crust associate a local peak of strain rate to an earthquake. These peaks are typically lasting a fraction of a second, $10^{-2} - 10^{-1}$ s, which corresponds to $10^3 - 10^4$ s for the real Earth. Hence, the time resolution does not allow us to detect the analogues of seismic waves in the form of high frequency acoustic emissions. For this, we would need to increase the time resolution of our sampling to microseconds and also develop local probes in-situ due to the absorption of these waves (recall the scaling law according to which 10 micron in the sample corresponds to 100 km, hence the need for the analogues of seismic stations within the sample). We thus agree that difficulties in accessing the acoustic emission is a limitation of our experimental set-up but we suggest that it is largely compensated by the good scaling match and the wealth of novel results we obtain.

- The application of a controlled shear stress to the sample contrast also with the real nature of a tectonic fault, which is driven at a relative constant shear velocity.

>> The referee raises a good point but it would be fair to state that nobody knows the “real” nature of the driving of a tectonic fault or region. By geodetic methods, one measures strain rates at the surface of the crust and, by borehole methods and others, one can have access to the local stress field at specific locations. The “real” nature of the driving of a tectonic fault is likely more complicated than just a pure constant shear rate, as it depends on complex mantle convection configurations, interactions at plate boundaries, rheological conditions at the interface between the lower crust and the mantle, the thermal-visco-elastic strength of the lower crust, and more generally on the partitioning of the elastic strength of the various elements of the Earth crust and mantle. A constant strain rate requires an infinitely rigid (strong) driving plate. A constant stress rate requires an infinitely deformable driving spring. It is clear that the real crust lies in between. Hence, the controlled shear stress in our

experiments is not so artificial and provide a useful end-member of various driving conditions.

- Notice also the existence of analogue experiments with gels (Yamaguchi 2011).
- FX Passelègue, A Schubnel, S Nielsen, HS Bhat, R Madariaga, From sub-Rayleigh to supershear ruptures during stick-slip experiments on crustal rocks, *Science* 340 (6137), 1208 (2013).
- A Schubnel, F Brunet, N Hilairet, J Gasc, Y Wang, HW Green, Deep-focus earthquake analogs recorded at high pressure and temperature in the laboratory, *Science* 341 (6152), 1377 (2013).
- MM Scuderi, C Marone, E Tinti, G Di Stefano, C Collettini, Precursory changes in seismic velocity for the spectrum of earthquake failure modes, *Nature geoscience* 9 (9), 695 (2016).
- T Yamaguchi, M Morishita, M Doi, T Hori, H Sakaguchi, JP Ampuero, Gutenberg-Richter's law in sliding friction of gels, *Journal of Geophysical Research: Solid Earth* 116 (B12), (2011).

>> We thank the reviewer for bringing up these important points. We would like to clarify that our goal of this study is not limited to mimic earthquake-like behaviour in a system accessible in the laboratory but is much broader. There are quite a few important studies in recent years (as pointed out by the reviewer as well as many of our references), which aim to mimic earthquake dynamics in systems mechanically very similar to the earth-crust (like rock samples, system of hard grains, charcoal block etc.). The acoustic emissions from the sample in such experiments have been found to be very useful and successful to mimic earthquake statistics with a very wide range of energy and very good statistical accuracy due to high frequency data acquisition probing the failure events. Despite these significant efforts, the microscopic mechanism, giving rise to well-known scaling laws observed for foreshock and aftershock events, still remains to be fully understood. Our study aims to extend the idea of self-organization, observed in the broad realm of soft condensed matter, to the earthquake like dynamics. The main motivation behind such study stems from the fact that the length scales, time scales, effective interactions between the particles or domains, etc. in condensed matter systems can be tuned and probed, thus helping to further unravel the mechanisms behind earthquake statistics.

We would also like to point out that the constant shear stress induced quakes without volume contraction in our system show not just Gutenberg-Richter law, but also show Omori law as well as the correct scaling properties for the waiting time distribution (based on our new analysis). Despite of the wide differences in shear moduli of soft yield stress materials in our case (yield stress only few tens of Pa) and the earth crust, such remarkable statistical similarity (as established from three scaling laws) points to the fact that the microscopic origin triggering such events is general and not confined to materials having similar mechanical properties like the earth-crust.

Simultaneous observation of these three scaling laws has been reported for laboratory experiments with solid samples or collection of granular solid materials, however, for soft gels (similar to our case) there is no such experimental report. We also, provide direct experimental evidence of domain reorganization accompanying the burst phenomena in CTAT-nematic system.

Yamaguchi T, et al have observed only the Gutenberg-Richter's law in sliding friction of gels. Recently, using Lattice-Boltzmann simulation method for a compact system of bubbles, Benzi R, et al (Geophys. J. Int., 2016) have observed both the Gutenberg-Richter's law and Omori law below the yield stress with the exponents being close to those found for actual earthquakes. There is no experimental report with soft matter of few Pa yield stress proving these predictions. Our experiments closely resemble the mentioned simulations.

We have significantly rewritten the introduction to clarify these points and added suitable references in the revised manuscript.

2- Concerning one burst: "However, for $\sigma > 0.1$ Pa and beyond, a burst occurs at ~ 500 s that persists for ~ 1000 s." (on figure 1b,c)

- It is not clearly written in the manuscript what is the equivalent of this burst in the case of real earthquakes. It took me a second reading to realise (by reading the main claims of the paper in the introduction) that one burst corresponds to a sequence foreshock-aftershocks. Am I right? If so, it would be nice to better describe the dynamics: e.g. how many burst, and how different are they during one typical experiment?

>> We thank the reviewer for pointing out the confusion. As pointed by the reviewer, there is one main shock in one burst in Fig. 1(b) for $\sigma > 0.01$ Pa. The expanded version of the data for $\sigma = 2$ Pa in the time window of 350 to 750 sec is now included in the supplement (Fig. S3) where τ_s (centre of the burst appearing in the Omori law) is also marked. In our case, a burst refers to a sequence of foreshocks and aftershocks. We term a single spike within a burst as a quake.

In CTAT system, we observe only a single burst that corresponds to a sequence of foreshock-aftershocks (e.g. Fig.1b, c, Fig.4a, Fig.S5). The occurrence time of the main shock shifts towards the later time for higher sample thicknesses. We have performed new experiments with laponite under different sample conditions. We show the data for repeated experimental runs on the same loading of the sample (Fig.S13), and the aging-time dependence (aging time t_{glass} is the time gap between the end of the pre-shear and the start of the creep run as shown in Fig.S14).

Reviewing our laponite data sets, we have observed both single and multiple burst event(s) over the entire duration of our experiments (Fig.5a, Fig.S12, Fig.S13, Fig.S14). In some cases, the multiple burst events being temporally very close, foreshocks and aftershocks for different bursts overlap. In such scenario, low energy cut-off effects become significant that results in a reduction of the range of energies over which Gutenberg-Richter's law can be verified (Fig.S11).

We have clarified these points in the revised manuscript.

- Figure 2 indicates a G-R relation in the “statistical properties of bursts”. To avoid the confusion, I think that it is important (if the authors decide to continue on exploiting the similitude with earthquakes) to give a name to their equivalent to “one earthquake”, which may be one spike within a burst. During my first reading I was considering that one burst was one earthquake and I was wondering how many events of 1000 s were analysed in the statistics?

>> We thank the reviewer for this very useful suggestion. We have now named a single spike within a burst as a quake, equivalent to one earthquake. Thus, within a burst we have many quake events. The number of quakes (N) considered for statistics is now explicitly mentioned in the revised manuscript (Fig.2b, Fig.5c).

3- If my analysis so far is correct, there is one mainshock in each burst, defining foreshocks to the left and aftershocks to the right of it. It would be interesting to analyse the statistics of mainshocks to see if they follow a G-R law, also it is important to indicate the origin of the spikes analysed in figure 2: From how many bursts? How different are those bursts? Are they from different experiments or they all are from the same experiment?

>> Following the reviewer’s suggestions, we have now clearly defined the burst events comprising of many spikes. Each spike we now call as a single quake.

In a single experimental run (over 8000 s) we only observe a single burst for CTAT (Fig. 1b and 1c). In case of laponite, we observe both single/multiple bursts for different experimental conditions (as shown in Fig.5a, Fig.S12, Fig.S13, and Fig.S14). However, even for the case of multiple bursts, the number of bursts is too few (e.g. three bursts in Fig.5a, Fig.S13) to carry out G-R statistics on bursts. We have clarified these points in the revised manuscript.

CTAT $\sigma = 2$ Pa, $d = 10$ micron, single burst data is considered to plot the G-R statistics in Fig.2a. Considering all experimental data sets of CTAT, the G-R plot is shown in Fig.S6. Note that the G-R statistics of all data for a given sample thickness roughly fall on a master curve. Following the referee’s suggestion, we now clearly indicate the number of events (corresponding to different energy cut-offs) considered for each analysis.

In case of laponite, we have also analysed multiple burst events. They display remarkable statistical similarities with the single burst case. The waiting time distribution (Fig.5c) and frequency distribution in time (Fig.S10) show power law behaviour with very similar exponents as observed in case of a single burst.

In Fig.S11, we show the distribution of magnitude of energy (G-R law), where the data show significant deviation due to the low energy cut off effects, most likely originating from the overlap of foreshocks and aftershocks for multiple bursts.

4- The exponent *Beta* is not constant, but varies from around 0.6 to more than 1.0, which is huge in terms of earthquakes (see Supp. Mat in ref. 5). Maybe there is some interesting physics behind. See for example (Barés 2018).

-J Barés, A Dubois, L Hattali, D Dalmas, D Bonamy, *Aftershock sequences and seismic-like organization of acoustic events produced by a single propagating crack*, *Nature communications* 9 (1), 1253 (2018).

>> We thank the referee for this interesting remark. There is indeed an active literature in statistical seismology dealing with the observation (or rebuttal) of variations of b-values and its possible significance in terms of providing a probe to the stress field in the Earth crust. See for instance the critical review in Kamer, Y., and S. Hiemer (2015), Data-driven spatial b value estimation with applications to California seismicity: To b or not to b, *J. Geophys. Res. Solid Earth*, 120, doi:10.1002/2014JB011510. There are also recent claims that a real-time variation of the b-value could be informative and even predictive of the subsequent nature of the triggered earthquakes, see L. Gulia and S. Wiemer, Real-time discrimination of earthquake foreshocks and aftershocks, *Nature* 574, 193 (2019). However, these works and others raise many questions about their reliability and we would rather avoid entering this debate here, in order to focus on the main results of our experiments.

5- *Why the need of wavelets analysis to introduce an Omori law, which is quite easy to analyse in its original form? Presenting the original way (if possible) must be a stronger indication of the similarity to actual earthquakes.*

>> Following the reviewer's suggestion, we have done fresh Omori-law analysis with CTAT $\sigma=2$ Pa, $d=10$ micron data. We now directly plot the rate of quakes around the main-shock as a function of time t' (Fig.3b in the revised manuscript). We have fitted the Omori-Utsu power law $N = N_0 + A / (c + t')^p$ with $p = 1$ as also used in other experiments (e.g., Lherminier S. et al., PRL, 2019). The Omori exponent ~ 1 is also obtained using the wavelet analysis (Fig.3a).

The main advantages of using wavelets over rate count are that such analysis does not require the low energy cut-off for background activity and suitable time window of binning to get good statistics. Further, it reveals the log-periodic variation as pointed out by us in the 'discussion' section, which indicates a discrete scale invariance of wavelet coefficients near the singularity. It would be interesting to follow up on this aspect in more experiments in future.

Reviewing on the dynamics: The foreshocks/aftershocks sequences are not similar to the ones presented in real earthquakes. In nature there is a very important asymmetry between foreshock and aftershocks activities (very few foreshocks

against a very large number of aftershocks). In this manuscript we find the opposite asymmetry with a large and progressive activity in the foreshock zone.

>> We thank the reviewer for this important point. It is interesting to note that apart for a few exceptions, the real earthquakes show this interesting asymmetry between foreshock and aftershock events. In our case, CID analysis of polarizing optical microscopy images (for CTAT sample) reveals that, in the foreshock regime, rapid coarsening of nematic domains takes place (Fig.4g and 4h). Such stress induced domain reorganizations drive the system to a much stronger solid-like state (Fig.S1). We further note that, for CTAT system, the duration of foreshock events depends strongly on system size (e.g. for larger gaps between the shearing plates, the foreshock duration increases). We believe that when the reorganized nematic domains becomes comparable to the size of the gap, the system gets jammed and reaches a steady state. Thus, the foreshock duration indicates a timescale for the domains to span the gap. In the revised version we briefly discuss these points.

The only similarity is the G-R law, which is an attribute of many different natural systems, so I do not consider that this system presents an earthquake-like dynamics that make it of interest for a community working on earthquake physics.

>> As already discussed above, with the new figures and modifications in the revised manuscript, the similarities with the three statistical scaling laws (G-R law, Omori law, waiting time distribution) are very clearly presented.

We would like to point out that our study is not just an extension of studying earthquake dynamics in another system. Our experiments reveal that the earthquake-like dynamics is intimately linked to the stress induced organization of nematic domains for CTAT system. We observe that the spatial auto-correlation of intensity ($A(k, t)$, in wave vector-space) decays at lower values of k as the time increases, which is a direct measure of the coarsening of the nematic domains. The bust like event with many foreshocks and aftershocks takes place during such domain coarsening dynamics.

Furthermore, experimental verification of Omori law as well as waiting time distribution with the power-law are being reported for the first time in gel-like soft matter systems.

Reviewer #3 (Remarks to the Author):

The aim of the work is to mimick earthquake-like statistics (Gutenberg-Richter energy-release distribution, Omori aftershock time-decay) in laboratory experiments. The authors refer to earlier experiments with such statistics on hard materials (rock, ceramic, wood) where earthquake-like statistics has been analyzed with acoustic emission (AE). They correctly argue that true comparison requires soft model

systems, for which AE cannot be used. Their choice is two soft yield-stress fluids, which they analyze below the yield stress with shear rheology and polarized optical microscopy. The observed statistics of energy bursts under shear indeed corresponds with the Gutenberg-Richter (GR) law, with exponent $b = 0.75$; the temporal behavior nicely mimicks Omori's law of aftershocks, with creep exponent α around unity. In both cases the scaling exponents are close to what is seen for earthquakes (b and α both around unity). In the interpretation the authors propose a picture of intermittent formation and breakup of bridging clusters of the colloidal particles across the rheometer gap. In general the goals are very relevant and this work seems well worth publishing.

>> We thank the referee for finding our work relevant & worth publishing after due revision.

There are however a number of objections or questions. First of all the authors state that "to our knowledge there is no similar study on soft and disordered materials that show solid-like yield stress". There is earlier work in soft systems that have been measured by other ways than AE, and that is very relevant in the context. Although these soft systems concern not always jamming yield-stress fluids they too show the same earthquake-like statistics, e.g.:

- Sprakel et al., *Phys. Rev. E* 79, 056306 (2009) already report intermittent dynamics with power-law stress-drop statistics in experiments of soft transient networks under shear; they propose a similar picture of fracturing and restoring network bonds. In this case the system is fluid at zero shear and has a soft stress peak in the flow curve; it causes stress fluctuations between a high- and low-viscosity state, which shows that jamming bridges across the full gap and true stick-slip are not necessary to give earthquake-like behavior: the system may still be a high-viscosity fluid at very long (e.g. geological) timescales.

- Zargar et al. have reported Gutenberg-like statistics in soft colloidal glasses: *Phys. Rev. Lett.* 110, 258301 (2013)

- Yamaguchi et al., *J. Geophys. Research* 116, B12306 (2011) publish GR statistics in stick-slip sliding friction of a hard block across a soft gel plate.

- Siebenburger et. al., *Phys. Rev. Lett.* 108, 255701 (2012) have measured logarithmic creep ($\alpha = 1$) in the long-time regime for sheared colloidal glasses well below the yield stress.

In addition to these experimental studies, a Lattice-Boltzmann simulation by Benzi et al., *Geophys. J. Int.* 207, 1667 (2016) reveals the same GR and Omori statistics as above for plastic events in a soft glass.

These earlier papers need to be discussed in detail in order to assess what is really novel here, and what is not. Provided that also questions on other aspects of the present manuscript can be satisfactorily answered, the work may still be considered a valuable extension of existing studies, with new model systems and novel and in some respects more detailed ways of measurement and analysis, deserving

publication in a very good journal.

>> We thank the reviewer for this important remark about the existing literatures where stress/shear-rate fluctuations show earthquake like statistics for few other soft matter systems. We would like to highlight the fact that to the best of our knowledge, our study is the only one that obtains three important scaling laws, namely, G-R Law, Omori Law and waiting time power law distributions (with our new analysis), by direct strain measurements (without acoustic emission). Furthermore, for soft systems, our study points out, for the first time, the role of self-organization in the system giving rise to burst-like events observed in real earthquakes. We quantify this reorganization of disordered domains by analysing polarized optical microscopy images using the concept of computational information density (CID). These novel aspects of our study can shed light on microscopic mechanism behind the statistical behaviour of foreshock and aftershock events in actual earthquakes which remains unknown. Below we give the differences between our study and existing papers, individually.

(i) The stress fluctuations observed by Sparkel et al. (PRE, 2009) were also observed for other polymeric / micellar fluids (Fardin et al. PRL 2010, Majumdar and Sood, PRE, 2011, Ganapathy et al. PRE, 2008, Majumdar and Sood, PRE, 2014...). Although, shear induced the microscopic fracture and recovery can produce such stress fluctuations, later studies (mentioned above) revealed the importance of secondary flows (Taylor vortices) in curvilinear geometries originating from elastic instabilities. In particular, the strong correlation between the vortex dynamics and the dynamics of stress fluctuations indicate that the simple stick-slip picture is not enough. Although Sparkel et al. probed the stress fluctuations without AE (similar to our study), the absence of drastic system size dependence (Majumdar and Sood, PRE, 2011) and stress induced changes in elastic modulus due to domain reorganization (Fig.S1) makes this study very different from ours. They only probe G-R Law but not the Omori and waiting time power-law distributions.

(ii) The paper by Zargar et al. reports G-R like statistics in microscopic rearrangements of hard sphere colloidal glass. This system is very similar to ours (particularly laponite), but there is a significant difference in the range of interaction between the particles. The long range coulomb interaction between the laponite platelets makes them very different from a hard sphere system.

Moreover, the shear rate fluctuations in our case are not thermal and they show up in spatially averaged measurements like bulk rheology under an applied stress. The effective temperature indicating the energy scales of these fluctuations is $\sim 10^{11} - 10^{12}$ K (majumdar and sood, PRL, 2008; majumdar and sood, PRE, 2012). Thus, our study is very different from that by Zargar et al. However, it will be very interesting to study the non-equilibrium fluctuations in hard sphere colloidal glass

under external driving. They also only probe G-R Law and not the Omori and waiting time power-law distributions.

(iii) The study by Yamaguchi et al. study the stick-slip behaviour of frictional interfaces formed by soft and hard surfaces. Effectively it is a study of 2-D system where a nice correlation between stress drops and effective contact area has been reported. Although, the study is relevant, it does not give any indication about microscopic precursor (like domain coarsening in our case) leading to the failure events. They also did not probe the Omori-law.

(iv) The creep measurement by Siebenburger et. al. is indeed very relevant for our study. It will be interesting to analyse the fluctuations they observe (adjusting the system size and sampling rate suitably) below the fluidized state to see whether or not such similar burst and G-R and Omori statistics come in the picture.

All these studies measure the sample response directly (like our case) and not with AE. However, no such experimental study on soft materials has probed Omori scaling laws. Stress induced reorganization in our system give rise to persistent foreshocks over ~ 500 s which has never been reported.

The Lattice-Boltzmann simulation study by Benzi et al. on a compact system of bubbles is indeed very relevant to our study. This paper mentions that the events associated with the reorganization of the system under a constant strain rate similar to real earthquake foreshocks and aftershocks. They also discuss both G-R as well as Omori scaling laws.

We have now added a discussion about these soft matter systems with suitable references in the revised manuscript.

I also have the following questions:

- The authors suggest that their experiments, in contrast to those on the hard systems, favorably compare with earthquakes a.o. because of the similar absolute value of the sampling rate, measured in Hz. Should a true correspondence between strongly differing systems such as geological faults and colloids not rather be based on comparison of dimensionless timescales and stresses, using characteristic times and energies of each system in the reduction? Can the authors provide such a scaling?

>> The referee is right and this was missing in the previous version. We have now added a full derivation of the scaling correspondence between our experimental system and the seismogenic Earth crust. In particular, we show that, given the scaling ratio of length and time scales, the mapping between the two systems holds

for other physical quantities such as stress and shear modulus, strain rate and velocity of rupture as explained below and now added in the revised manuscript.

Modelling large scale phenomena such as tectonic events in the laboratory requires to scale properly both the loading stresses and the physical properties of the material. In order to establish the correspondence between the laboratory model and geological scales, we need to introduce the scaling factors $L_{sf} = L/L^*$ and $T_{sf} = T/T^*$ relating the length L^* and time scales T^* of the model to those, L and T , in the Earth crust. The spatial range over which tectonic shear stresses occur in the Earth crust due to tectonic motions is of the order of 100 km. The width of the gap over which the shear stress is applied in the experiment is ~ 10 micron. Thus $L_{sf} \sim 10^{10}$. The typical duration of earthquake clusters (including foreshocks-mainshock-aftershocks) is of the order of months to years. The typical time span of a burst in the laboratory experiment is about 1000 s, thus we have $T_{sf} \sim 10^5$. From these two scaling ratios $L_{sf} \sim 10^{10}$ and $T_{sf} \sim 10^5$, we proceed to derive the dimensional scaling laws of all other relevant physical quantities, demonstrating that our laboratory experiment is a good analog to the Earth crust.

An earthquake in the crust is associated with the driving shear stress overpassing the frictional force along faults. The frictional force is equal to the normal stress, controlled by the lithospheric pressure, times the friction coefficient, in the range 0.1-0.6. Hence, the scale of the stresses at which fault sliding is triggered is governed by ρgh , where ρ is the average density of the Earth crust, g is the acceleration of gravity and h is the width of the seismogenic crust. As ρ is about 3 times that of our suspensions, and g is the same, this predicts that the shear modulus in our experiments should be about $1/L_{sf}$ that of the Earth crust (roughly 10^{10} to 10^{12} Pa), i.e. 1 to 100 Pa in the experiments, which is of the same order as in our laboratory observations. The typical rupture sliding velocity during an earthquake is ~ 1 m/s, which should be $L_{sf} / T_{sf} \sim 10^5$ times the velocity of local peaks of the burst in our experiments. Indeed, the typical velocity of the shear motion at a local peak is $\sim R d\phi/dt \sim 1 \text{ cm} \times 1 \text{ mrad/s} = 10^{-5}$ m/s. Lastly, since strain is dimensionless, the strain rate of $\sim 0.1 \text{ s}^{-1}$ in our sample at a peak during a burst should correspond to $T_{sf} = 10^5$ times the strain rate during an earthquake, which is typically 1 m of slip over a fault length of 100 km occurring in 10 s of slip, which yields the strain rate $\sim 10^{-6}$, which is in perfect agreement with the prediction from the scaling laws.

- The Laponite system is introduced to ascertain the conclusions on the nematic system. But the former is aging at rest. To what extent do the results change with changing preparation protocol? To what extent may the timescale of aging interfere with the timescale of bursts? To what extent can the aging system actually be compared with the Herschel-Bulkley nematic system and can the same interpretive picture be used?

>> Guided by these comments, we have performed new experiments with laponite. In Fig.S12, we show the data for different sample concentrations and find that the G-R slope remains almost unchanged ($\epsilon \sim 1.5 \pm 0.1$). In Fig.S14, we show the

aging time dependence of shear-rate fluctuations and find that the temporal position of the bursts shifts systematically towards shorter timescales. In Fig.S13, we show that, for laponite system also, we can simply erase the memory by applying the pre-shear protocol. We agree with the reviewer that the laponite system is much more complicated due to the intrinsic aging of the sample at rest. However, we would like to point out that, even for this system, when the aging is sufficient to drive the system into a glassy state, the stress induced yielding shows approximately H-B behaviour (Fig.1, Majumdar and sood, PRL, 2008).

We have added these points in the revised manuscript.

- Can the authors give more quantitative information on the size of the dynamically coherent domains compared to the gap size.

>> Please refer to Fig.5. We see that when the average size of the nematic domains (~70 micron) becomes comparable to the size of the gap (150 um), the system gets jammed. We now explicitly mention this in the revised manuscript.

- Dinkgreve et al. Journal of non-Newtonian fluid mechanics 238, 233-241 (2016) perform similar creep experiments on a similar soft system, but do not see any intermittency. Do the authors have any idea what is specific to their systems, and what is generic?

>>There are four key parameters along with the choice of the sample: (1) the applied stress, (2) the sample thickness, (3) the data sampling frequency and (4) the creep measurement duration (Fig.1). There are many different creep studies with different types of samples, but the intermittency may be missed due to any one of the above key parameters. Dinkgreve et al. used cone-plate geometry (average sample thickness ~ 200 micron) to do creep measurements (sampling rate 2 Hz and measurement duration ~ 500 sec) above and below the yield stress region on carbopol gels, hair gel, emulsions and foam. It is possible that, due to the large sample thickness and low measurement duration, they might have missed the intermittency. We hope that our experiments will motivate new experiments to explore the flow of materials under creep conditions.

REVIEWERS' COMMENTS:

Reviewer #1 (Remarks to the Author):

The paper is most interesting and should be published

Reviewer #2 (Remarks to the Author):

Dear Editor,

In the nineties there were many papers motivated by the ideas of the "Self-organized criticality" reporting earthquake-like behavior. Unfortunately the resemblance of these systems with the real phenomenon was actually poor. However, besides the many different scaling and exponent values reported in numerical and experimental model systems, all they agreed (from a physicist perspective) with the "laws" of earthquakes.

As a result we find (still today) a huge collection of exponent values as "the one" of the Gutenberg-Richter law, even with negative b values [e.g. of one published recently in Nat. Com: Aftershock sequences and seismic-like organization of acoustic events produced by a single propagating crack, Nat. Com. 9, 1253 (2018)]. Notice that a negative b value means that large magnitude events are more often than small magnitude ones.

The inter-event time distribution of real earthquakes (according this work) is mean to behave as a power law decay with an exponent 1 (citing Bak P, et al. PRL, 2002). I can continue indefinitely listing misleading earthquake behavior reported mainly by physicists. At the early nineties maybe we had to rely on bad statistics and reports of earlier studies. However, this is no longer the case: today earthquake catalogs are rich in data and of easy access, so we cannot accept any longer references of earthquake behavior that not correspond to their reality. Analyzing real earthquake data (time series of magnitude of events) is much simpler that performing and analyzing devoted experiments, so my advice to the physicists trying to help improving the knowledge of earthquake physics is first downloading and treating real earthquake data.

- If one does that, one will find that the Gutenberg-Richter law has a b-value around $1 \pm 0,1$. Of course this value is obtained if the amount of data is considerable, with partial or incomplete data we can get whatever we want.

- In real earthquakes, the interevent distribution is not a power law decay with an exponent 1 (as mentioned in the manuscript), but a Gamma function containing an exponential decay for interevent times larger than a characteristic time (the inverse of the rate) and a power law with a very low exponent value (around 0.3) for interevent times smaller than this characteristic time (see Corral PRL-2004).

-Real foreshocks-aftershocks sequences are highly asymmetrical, with just a few events before the main shock, and a very large number of aftershocks; and always the activity is more important in the aftershock part (find a real sequence here: [https://en.wikipedia.org/wiki/Aftershock#/media/File:2016_Central_Italy_earthquake_\(magnitude\).svg](https://en.wikipedia.org/wiki/Aftershock#/media/File:2016_Central_Italy_earthquake_(magnitude).svg)). This is quite far from the foreshocks-aftershocks sequences presented in the manuscript, where there is always a larger activity in the foreshock part (figures 1b,c, 4a). This clear fact is, however, in contradiction with the new figure 3b showing that aftershocks are two or three times more abundant than foreshocks.

- In real earthquake sequences, there is a background activity plus aftershocks sequences on top of it. In this article it seems that there is no detectable background activity, but normally just one very important increase of the activity, which reaches a maximum and then decreases much faster than the former increase.

I am still convinced that the only similarity of the manuscript with the statistics of earthquakes is having the energy of events following a power law with an exponent value similar to the G-R one. But given the fact that power law distributed events are quite common in nature, I do not consider that the behaviors of these systems are "earthquake-like". The rest of the dynamics is quite different from the real behavior of earthquakes.

The phrase "at these large scales, the elastic moduli become small compared with gravity and the Earth crust should be considered as an effective very weak soft medium" is misleading: as a response, the authors use scaling arguments to indicate that the deformations values they have in their experiments are compatible with the ones observed in real earthquakes. However, this does not justify the quoted statement, neither the fact that very soft materials may be a good analogous of a tectonic fault.

I recommend not publishing this article in Nature Communications, and in general not publishing it if presenting its earthquake-like dynamics as its main result, which I consider misleading.

Reviewer #3 (Remarks to the Author):

The authors did a tremendous job in answering/addressing all the questions/comments of all the referees. I recommend publication as it is.

REVIEWERS' COMMENTS:

We thank the reviewers for examining our work. We appreciate that their detailed remarks and suggestions have helped us to improve our manuscript.

Reviewer #1 (Remarks to the Author):

The paper is most interesting and should be published

>> We thank the reviewer for finding our work interesting and recommending it for publication.

Reviewer #2 (Remarks to the Author):

- If one does that, one will find that the Gutenberg-Richter law has a b-value around $1 \pm 0,1$. Of course this value is obtained if the amount of data is considerable, with partial or incomplete data we can get whatever we want.

>> The argument by the referee #2 does not invalidate our mapping. Indeed, the logic is not that the energy power law (Gutenberg-Richter distribution) is uniquely characteristic of earthquakes. Rather, the energy power law is a kind of universal or ubiquitous signature of out-of-equilibrium slowly driven systems with threshold dynamics. The Earth crust qualifies as an out-of-equilibrium slowly driven systems with threshold dynamics and exhibits this power law.

As correctly described by the Referee, the 1990s (and even the 2000s) were characterised by cohorts of physicists claiming relevance of their simple models based on the approximate reproduction of the Gutenberg-Richter distribution. Two of the authors of this manuscript (Ouillon and Sornette) involved in the data analysis and modelling of earthquakes for decades have been long aware of this deficiency and have been among those few pushing for the documentation of several statistical laws in order for a mapping to be supported (Gutenberg-Richter law, Omori law and inverse-Omori law, productivity law, time and space fractal clustering of seismicity, and so on). This is what we have been aiming at in our present manuscript. We emphasize that the objective of our work is to draw the correspondence with the statistical analysis of the earthquake data and we show that it is sufficient strong at the semi-quantitative level to warrant the hypothesis that we propose in the manuscript.

- In real earthquakes, the interevent distribution is not a power law decay with an exponent 1 (as mentioned in the manuscript), but a Gamma function containing an exponential decay for interevent times larger than a characteristic time (the inverse of the rate) and a power law with a very low exponent value (around 0.3) for interevent times smaller than this characteristic time (see Corral PRL-2004).

>> One of the authors of this manuscript together with A. Saichev [1,2] has shown that this characterisation of the interevent distribution by Corral is kind-of-naive and that a much more general and parsimonious theory of the interevent distribution derives simply from the most powerful statistical model of seismicity, namely the ETAS model (which is an extension of the self-excited conditional point process introduced by Hawkes in 1971 and applied to earthquakes by Ogata in 1988 in the first version).

We stress that the theory of the interevent distribution in [1,2], which accounts remarkably well for the empirical data, does identify several power law regimes and intermediate asymptotics. Hence, our empirical finding of a power law interevent time distribution, while not reproducing the exact distribution for earthquakes, belongs to the same class.

[1] A. Saichev and D. Sornette,
"Universal" Distribution of Inter-Earthquake Times Explained,
Phys. Rev. Letts. 97, 078501 (2006)

[2] A. Saichev and D. Sornette,
Theory of Earthquake Recurrence Times,
J. Geophys. Res., 112, B04313, doi:10.1029/2006JB004536 (2007).

-Real foreshocks-aftershocks sequences are highly asymmetrical, with just a few events before the main shock, and a very large number of aftershocks; and always the activity is more important in the aftershock part (find a real sequence here: [https://en.wikipedia.org/wiki/Aftershock#/media/File:2016_Central_Italy_earthquake_\(magnitude\).svg](https://en.wikipedia.org/wiki/Aftershock#/media/File:2016_Central_Italy_earthquake_(magnitude).svg)). This is quite far from the foreshocks-aftershocks sequences presented in the manuscript, where there is always a larger activity in the foreshock part (figures 1b,c, 4a). This clear fact is, however, in contradiction with the new figure 3b showing that aftershocks are two or three times more abundant than foreshocks.

>> Following the reviewer's suggestion in the earlier report, we have plotted the rate of quakes around the main-shock (N) as a function of time t' (Figure 3b) having different background energy cut-offs: foreshocks $E_{\text{cutoff}} = 2.31 \times 10^{-1}$ in units of $\text{mrad}^2 \text{s}^{-2}$, aftershocks $E_{\text{cutoff}} = 2.57 \times 10^{-2}$ in units of $\text{mrad}^2 \text{s}^{-2}$ (already mentioned in the caption of the figure 3b). It is clear from Figure 3b that the background amplitude during the foreshocks is greater than the one during aftershocks (As E_{cutoff} is proportional to the square of the background amplitude; here foreshocks $E_{\text{cutoff}} >$ aftershocks E_{cutoff}). After the background subtraction, the rate of foreshocks is lower than the rate of aftershocks similar to the rate asymmetry of real earthquakes.

Except for Figure 3b, we have used the wavelets analysis throughout the paper where we have not subtracted any background to the data as the wavelet analysis does it automatically.

We have highlighted this similarity of the rate asymmetry with the real earthquakes in the modified manuscript.

The phrase “at these large scales, the elastic moduli become small compared with gravity and the Earth crust should be considered as an effective very weak soft medium” is misleading: as a response, the authors use scaling arguments to indicate that the deformations values they have in their experiments are compatible with the ones observed in real earthquakes. However, this does not justify the quoted statement, neither the fact that very soft materials may be a good analogous of a tectonic fault.

>> To avoid any misinterpretation, we have now deleted the line mentioned by the Reviewer:” At these large scales, the elastic moduli become small compared with gravity and the Earth crust should be considered as an effective very weak soft medium.”

I recommend not publishing this article in Nature Communications, and in general not publishing it if presenting its earthquake-like dynamics as its main result, which I consider misleading.

Reviewer #3 (Remarks to the Author):

The authors did a tremendous job in answering/adressing all the questions/comments of all the referees. I recommend publication as it is.

>> Thanks to the reviewer for crediting us for the job and recommending it for publication.